# A Case-Matched Analysis of Laparoscopic Liver Resection for Hepatocellular Carcinoma Located in Posterosuperior Segments of the Liver According to Adaption of Developed Techniques

**DOI:** 10.3390/medicina58040543

**Published:** 2022-04-14

**Authors:** Yujin Kwon, Boram Lee, Jai Young Cho, Ho-Seong Han, Yoo-Seok Yoon, Hae Won Lee, Jun Suh Lee, Munwhan Kim, Youngsoo Jo

**Affiliations:** 1Department of Surgery, Seoul Medical Center, Seoul 02053, Korea; nesia7979@gmail.com; 2Department of Surgery, Seoul National University Bundang Hospital, Seoul National University College of Medicine, Seoul 13620, Korea; boramsnubhgs@gmail.com (B.L.); hanhs@snubh.org (H.-S.H.); yoonys@snubh.org (Y.-S.Y.); lansh@hanmail.net (H.W.L.); rudestock@gmail.com (J.S.L.); imedi06@snubh.org (M.K.); 82588@snubh.org (Y.J.)

**Keywords:** hepatectomy, prognosis, complication, survival

## Abstract

*Background and Objectives*: Laparoscopic liver resection (LLR) for the hepatocellular carcinoma (HCC) located in posterosuperior (PS) segment is technically demanding, but has been overcome by accumulated experiences and technological improvements. We analyzed peri-and post-operative results before and after the adaptation of the enhanced techniques. *Materials and Methods*: We retrospectively reviewed 246 patients who underwent LLR for HCC in PS segments from September 2003 to December 2019. According to the introduction of advanced techniques including intercostal trocars, Pringle maneuver, and semi-lateral French position, the patients were divided into Group 1 (*n* = 43), who underwent LLR from September 2003 to December 2011, and Group 2 (*n* = 203), who underwent LLR from January 2012 to December 2019. Among these cases, 136 patients (Group 1 = 34, Group 2 = 102) were selected by case-matched analysis using perioperative variables. *Results:* Mean operation time (362 min vs. 291 min) and hospital stay (11 days vs. 8 days, *p* = 0.023) were significantly longer in Group 1 than Group 2. Otherwise, disease-free survival (DFS) rate was shorter and resection margin (1.3 mm vs. 0.7 mm, *p* = 0.034) were smaller in Group 2 than Group 1. However, there was no difference in type of complication (*p* = 0.084), severity of complication graded by the Clavien–Dindo grade system (*p* = 0.394), and 5-year overall survival (OS) rates (*p* = 0.986). In case-matched analysis, operation time (359 min vs. 266 min *p* = 0.002) and hospital stay (11.5 days vs. 8.0 days, *p* = 0.032) were significantly different, but there was no significant difference in resection margin, DFS, and OS. *Conclusions*: The adaptation of improved techniques has reduced the complexity of LLR in PS segments.

## 1. Introduction

Laparoscopic liver resection (LLR) has been widely applied due to its advantage of minimal invasiveness resulting in earlier recovery and shorter hospital stay than open surgeries. It was initially limited to benign cases with easily accessible location, but its indication has been expanded to malignancy cases including hepatocellular carcinoma (HCC) and liver metastasis of other primary malignancies. Initially, LLR was only recommended for hepatic lesions less than 5 cm, solitary and located in anterolateral segments (AL segments 2, 3, 5, 6 and the inferior part of segment 4) and the lesions in left lateral section, which was considered to be a standard procedure [1]. However, due to the risk and the technical difficulty of approach and transaction of the liver parenchyma, open liver resection (OLR) has been favored for large tumors and lesions located in posterosuperior segments (PS segments, which demonstrated the segments 1, 7, 8 and superior part of segment 4) [2,3].

Owing to advanced laparoscopic operative techniques, instruments and accumulated experiences, these limitations have been overcome and in the Southampton consensus guidelines in 2017 acknowledge that LLR for lesions in PS segments are feasible and safe and should be considered as a valid alternative approach in expert centers [4]. There are many feasibility studies, but comparative studies are rare. Therefore, perioperative characteristics and operative results before and after the adaptation of developed techniques were compared in this retrospective review.

## 2. Materials and Methods

### 2.1. Patients

We analyzed the medical records of 246 patients who underwent LLR for HCC in PS segments at Seoul National University Bundang Hospital (Seongnam, Korea) between September 2003 and December 2019. According to the introduction of advanced techniques including intercostal trocars, Pringle maneuver, and semi-lateral French position, the patients were divided into Group 1 (*n* = 43), who underwent LLR from September 2003 to December 2011, and Group 2 (*n* = 203), who underwent LLR from January 2012 to December 2019. To reduce potential bias attributed to the retrospective study design, the patients in both groups were matched on 1:3 basis for the following preoperative variables: age, sex, BMI, tumor size, and Child-Pugh class. Finally, a total of 136 patients were selected, divided into Group 1 (*n* = 34, 25%) and Group 2 (*n* = 102, 75%).

### 2.2. Definitions

Parenchymal-sparing liver resection was performed whenever possible, and patients were all pathologically confirmed to have HCC. If the patients had multiple tumors, after multidisciplinary discussion, patients underwent preoperative transarterial chemoembolization (TACE) or radiofrequency ablation preoperatively [5]. All patients with cirrhosis were pathologically confirmed [6]. Additionally, select patients with large solitary HCC underwent LLR [7] and informed consent was obtained from all patients before the operation.

Complications were categorized to general, surgical, and liver-related, and mixed and major complications were defined as complications of Grade III or above, according to the Clavien-Dindo classification [8]. This study was approved by our institutional review board (B-2003-600-104).

### 2.3. Surgical Techniques

The surgical techniques used for LLR in our hospital have been described [3,8,9,10]. Under general anesthesia, the patient was tilted to 30° reverse Trendelenburg position with lithotomy and was placed right-side-up or left lateral decubitus adjustment with legs spread (French position). A 12 mm camera port was placed at the subumbilical region followed by two 11- or 12 mm main working ports inserted into the epigastrium and right upper quadrant of the abdomen along the subcostal area. Subsequently, 5 mm ports were placed in the left subcostal area for the assistant and two additional ports were placed at the 7th and 9th intercostals spaces with caution to prevent intercostal vessel bleeding. The surgeon stood between the patient’s legs at the beginning and moved to the patient’s right side to operate via intercostal trocars. The endoscopist and assistant stood on the left side of the patient. Flexible intraoperative ultrasonography was performed to localize exact tumor region and confirm adjacent vasculature to maintain the appropriate resection margin. Then, to minimize bleeding during parenchymal transaction, Pringle’s maneuver was applied; after dissecting hepatoduodenal ligament, it was encircled with umbilical tape. Both ends of the tape were passed through the long tube to apply intermittent clamping. To avoid the ischemic damage to the parenchyma, each clamping did not exceed 15 min. Next, superficial hepatic parenchyma was transected with ultrasonic shears, whereas deeper parenchyma was transected with laparoscopic Cavitron Ultrasonic Surgical Aspirator (CUSA, Integra Lifesciences, Plainsboro, NJ, USA). Bleeding from small branches of the hepatic veins was controlled with endo clips and a sealing device. The resected specimen was inserted into a protective bag and retrieved through the extended trocar site and if the specimen was large, extra transverse incision was made to suprapubic area. Finally, irrigation and hemostasis were performed followed by fibrin glue application to the transaction plane and the wound was closed in layers.

### 2.4. Statistics

Continuous variables were compared using independent sample *t*-test and presented as mean ± standard deviation. Categorical variables were compared using Fisher’s exact test. Survival rates were calculated using the Kaplan–Meier and life table methods and the differences between the groups were assessed using a log-rank test. All analyses were performed using SPSS version 19.0 for Windows (SPSS Inc., Chicago, IL, USA) and the differences were considered significant at *p*-values of <0.05.

## 3. Results

### 3.1. Characteristics and Outcomes in All Patients

During the period of study, 246 patients underwent LLR for HCC located in PS segments. They were divided into Group 1 (*n* = 43) and Group 2 (*n* = 203), and preoperative characteristics are summarized in Table 1. There were significant differences between the two groups in regard to age (55 vs. 62, *p* < 0.001), gender (*p* = 0.029) and presence of hepatitis (*p* = 0.022). However, body mass index, tumor size, indocyanine green retention rate at 15 min (ICG-R15%), and alpha-fetoprotein level indicated no significant difference. Additionally, the number of patients who underwent preoperative TACE or RFA was similar between groups. There were significant differences in most frequently performed operation type and operation time between groups (*p* = 0.031). The most commonly performed operation in Group 1 was right posterior sectionectomy (*n* = 11, 25.6%), whereas tumorectomy was mostly performed in Group 2 (*n* = 76, 37.4%). The operation time was significantly shorter in Group 2 (362 min vs. 291 min, *p* = 0.009). On the contrary, there was no difference between open conversion rate (18.6% vs. 13.8%, *p* = 0.475), estimated blood loss (1034 mL vs. 908 mL, *p* = 0.609), or number of transfusion (30.3% vs. 24.1%, *p* = 0.441) between groups (Table 2). Regarding pathologic results, resection margin was greater (1.3 vs. 0.7, *p* = 0.034) and there were more patients with cirrhosis (74.4% vs. 55.2%, *p* = 0.026) in Group 1. However, there were no significant differences in other variables including presence of satellite nodules (14% vs. 7.4%, *p* = 0.224), microvascular invasion (41.9% vs. 44.8%, *p* = 0.739) and rate of R0 resection (97.7% vs. 94.6%, *p* = 0.739). Additionally, there was no difference in type (*p* = 0.084) or severity (*p* = 0.394) of complication according to Clavien-Dindo grade [11]. In addition, hospital stay was significantly shorter in Group 2 (11 days vs. 8 days, *p* = 0.023), but there was no significant difference in rate of recurrence (60.5% vs. 43.8%, *p* = 0.064) or recurrence pattern (*p* = 0.168).

### 3.2. Characteristics and Outcomes in Matched Groups

The patients were matched by preoperative variables and were compared again. There were no differences between groups in preoperative characteristics (Table 1). Type of resection (*p* = 0.031) (Table 2), operation time (359 min vs. 266 min *p* = 0.002) and hospital stay (11.5 days vs. 8.0 days, *p* = 0.032) were significantly different after matching. However, there was no significant difference in resection margin (1.2 mm vs. 0.7 mm, *p* = 0.075) (Table 3), DFS or OS in two groups after matching (*p* = 0.143) (Figure 1 and Figure 2).

## 4. Discussion

Laparoscopic liver resection for HCC in PS segment before and after the adaptation of advanced techniques was analyzed in this study. LLR for HCC has benefits in minimal invasiveness including smaller incision, earlier recovery, and shorter hospital stay [9,12,13] without oncological compromise [14,15].

In previous studies, peri-and postoperative results were compared [14,16]. Intraoperative bleeding, operative time, and hospital stay were significantly greater in PS group, but there was no significant difference in open conversion rate, OS, or DFS between groups. However, LLR in PS segment still had complications of parenchymal transection, bleeding control and visualization of deeper lesions. [2,3,13] Therefore, in the early era of LLR, it was only recommended for those patients with lesions located in AL segment (segment 2,3,5,6 and the inferior part of segment 4) or LLS, HCC less than 5 cm and solitary [1]. To overcome this limitations, numerous types of equipment and technologies were adapted.

Flexible scope enabled better visualization and use of ultrasonic shears, whereas laparoscopic CUSA and Bipolar Vessel Sealing System (Ligasure, Medtronic, Dublin, Ireland) minimized blood loss and facilitated parenchymal dissection. Additionally, by routine use of intraoperative ultrasound, delineation of adequate tumor-free margin was achieved. However, operating on HCC with a size of more than 5 cm and located in PS segment remained complicated, which was overcome by accumulated experience and adaptation of new techniques [3,9,16].

Tumors located in PS segment were limited to highly experienced surgeons owing to lack of space for manipulation of instruments and difficulty of transecting parenchyma due to blockage of the operative field [17]. Additionally, the laparoscopic equipment was too short for reaching the operative field, and therefore, hemostasis was difficult [13]. These limitations were overcome by adaptation of use of intercostal trocars, Pringle’s maneuver and modified semi-lateral position [8,15].

By adopting intercostal trocars, tumors located in unfavorable lesions such as the dome or posterior part of the liver were able to be operated on. By using Pringle’s maneuver, the amount of hemorrhage was decreased, resulting in a reduction in transfusion when blood loss is a prognostic factor of postoperative morbidity and mortality after liver resection [16]. Additionally, with decreased bleeding, a clear operative field was guaranteed and resulted in shortening of operative time. Moreover, by positioning patients in semi-lateral French position with 30-degree reverse Trandelenberg, the remnant liver was descended, facilitating approach. Additionally, by lifting the right hepatic vein above the inferior vena cava, venous bleeding was reduced [1,4,14,18].

Many times, HCC arises from cirrhotic liver with inadequate liver function [19] and therefore, conservation of liver parenchyma should be maximized with secure resection margin. Due to the difficulty of manipulation and hemostasis, nonanatomical resection in PS segment was reported as difficult and there were reports with inferiority of nonanatomical liver resection against anatomical resection although it is controversial [14,20,21]. However, it is important to balance adequate resection and remnant parenchyma and recently, there was a propensity score matching controlled study reporting no benefit in overall survival of patients who received nonanatomical resections as compared to patients with anatomical resections [22,23]. In addition, adaptation of advanced techniques not only facilitated the procedure, but also achieved better postoperative results.

### Limitation

There are several limitations. First, the retrospective, non-randomized nature of this study represents its biggest limitation. Second, due to the small sample size, it was not possible to match the type of operation. It may be confounding factors for results. Thirds, although, we divided the patients based on the changes of technical changes at our institutions, it slightly to be arbitrary.

## 5. Conclusions

Short-term outcomes of LLR for HCC in the PS segments have been significantly im-roved after the introduction of advanced techniques including intercostal trocars, pringle maneuver, and semi-lateral French position. Although the complexity of LLR in PS segments has been slightly reduced due to the development of theses improved techniques, further studies are needed.

## Figures and Tables

**Figure 1 medicina-58-00543-f001:**
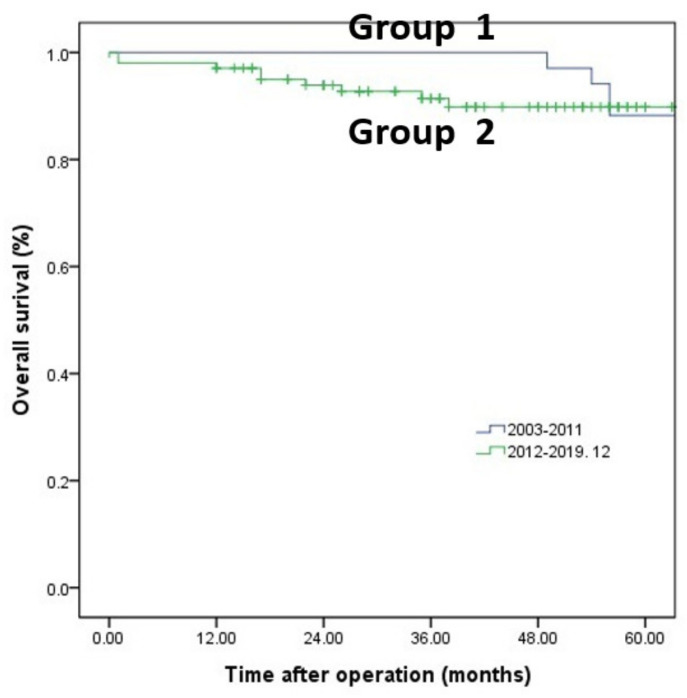
Overall survival: five-year overall survival rate in Group 1 and Group 2.

**Figure 2 medicina-58-00543-f002:**
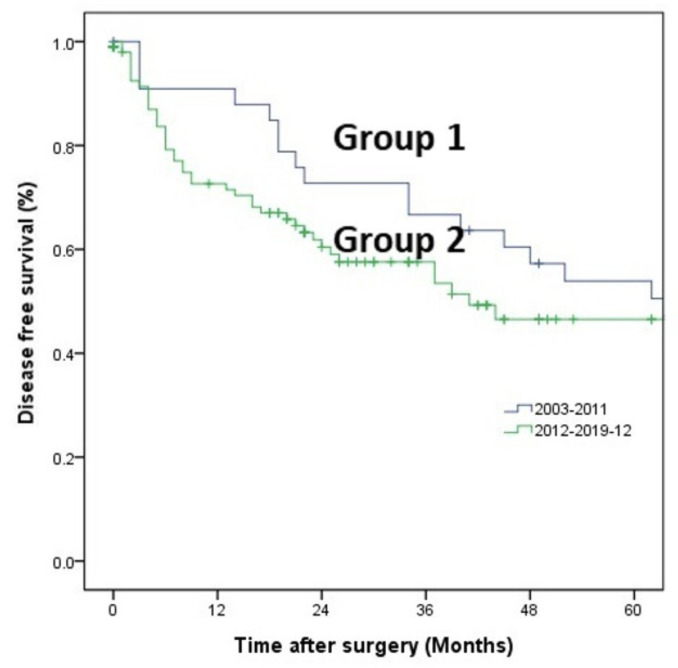
Disease free survival: five-year disease free survival rate in Group 1 and Group 2, case-matched analysis.

**Table 1 medicina-58-00543-t001:** Preoperative characteristics of laparoscopic liver resection for hepatocellular carcinoma in posterosuperior segments of the liver.

	All Patients			Case-Matched Patients		
Characteristics	Group 1 (*n* = 43)	Group 2 (*n* = 203)	*p* Value	* Group 1 (*n* = 34)	* Group 2 (*n* = 102)	*p* Value
Age	54.7 ± 9.4	61.6 ± 11.1	<0.001	56.1 ± 9.7	58.6 ± 10.3	0.202
Gender			0.029			1.000
Male	27 (62.8)	161 (79.3)		25 (73.5)	73 (71.6)	
Female	16 (37.2)	42 (20.7)		9 (26.5)	29 (28.4)	
* BMI (kg/m^2^)	23.8 ± 2.9	24.9 ± 3.5	0.062	24.0 ± 3.0	24.9 ± 3.4	0.151
Tumor size (cm)	3.2 ± 1.6	4.2 ± 7.5	0.415	3.0 ± 1.3	4.2 ± 10.2	0.493
* ICG-R15	17.9 ± 38.6	13.2 ± 12.1	0.420	19.6 ± 43.1	13.3 ± 13.3	0.479
* AFP (ng/mL)	563.4 ± 1777.4	631.6 ± 5016.8	0.930	344.8 ± 1240.7	518.9 ± 3584.6	0.782
Child-Pugh class, *n* (%)			1.000			0.680
A	41 (95.3)	191 (94.1)		33 (97.1)	96 (94.1)	
B	2 (4.7)	12 (5.9)		1 (2.9)	6 (5.9)	
C	0	0		0	0	
Hepatitis, *n* (%)			0.022			0.465
Hepatitis B	35 (81.4)	124 (61.4)		27 (79.4)	75 (73.5)	
Hepatitis C	3 (7.0)	12 (5.9)		3 (8.8)	6 (5.9)	
Both negative	5 (11.6)	66 (32.7)		4 (11.8)	21 (20.6)	
Prior * RFA, *n* (%)	4 (9.3)	14 (6.9)	0.529	3 (8.8)	9 (8.8)	0.618
Prior * TACE, *n* (%)	11 (25.6)	46 (22.7)	0.693	9 (26.5)	25 (24.5)	0.822

* Group 1 patients who underwent liver resection before introduction of advanced approach in 2012, group 2 patients who underwent liver resection after introduction of advanced approach in 2012, BMI body mass index, AFP alpha-fetoprotein, ICG-R15indocyanine green clearance rate at 15 min, RFA radiofrequency ablation, TACE transarterial chemo-embolization.

**Table 2 medicina-58-00543-t002:** Perioperative outcomes of laparoscopic liver resection for hepatocellular carcinoma in posterosuperior segments of the liver.

Characteristics	* Group 1 (*n* = 34)	* Group 2 (*n* = 102)	*p* Value
Operation type, *n* (%)			0.031
Tumorectomy	5 (14.7)	46 (45.1)	
Segmentectomy	8 (23.5)	16 (15.7)	
Bisegmentectomy	0	3 (2.9)	
Left hemihepatectomy	1 (2.9)	3 (1.5)	
Extended left hemihepatectomy	0	0	
Right anterior sectionectomy	1 (2.9)	6 (5.9)	
Right posterior sectionectomy	9 (26.5)	8 (7.8)	
Right hepatectomy	7 (20.6)	8 (7.8)	
Extended right hepatectomy	0	1 (1.0)	
Central bisectionectomy	1 (2.9)	1 (1.0)	
Combined resection	0	1 (1.0)	
Caudate lobectomy	2 (5.9%)	5 (4.9)	
Extended segmentectomy	0	4 (3.9)	
Extended right anterior sectionectomy	0	0	
Extended right posterior sectionectomy	0	1 (0.7)	
Type of resection, *n* (%)			0.017
Anatomical	23 (67.6)	44 (43.1)	
Non-anatomical	14 (32.6)	106 (51.5)	
Operation time (min)	359.4 ±143.6	265.58 ±150.8	0.002
Pringle method, *n* (%)	5 (14.7)	65 (63.7)	<0.001
Open conversion, *n* (%)	7 (20.6)	9 (8.8)	0.475
Blood loss (ml)	1103.2 ±2073.0	859.1 ±1287.2	0.418
Transfusion, *n* (%)	11 (32.4)	20 (19.6)	0.157

* Group 1 patients who underwent liver resection before introduction of advanced approach in 2012, group 2 patients who underwent liver resection after introduction of advanced approach in 2012.

**Table 3 medicina-58-00543-t003:** Pathologic and postoperative outcomes of laparoscopic liver resection for hepatocellular carcinoma in posterosuperior segments of the liver.

Characteristics	* Group 1 (*n* = 34)	* Group 2 (*n* = 102)	*p* Value
Resection margin (mm)	1.2 ±1.4	0.7 ±0.6	0.075
Cirrhosis, *n* (%)	24 (70.6)	64 (62.7)	0.535
Satellite nodules, *n* (%)	3 (8.8)	2 (2.0)	0.100
Microvascular invasion, *n* (%)	13 (39.4)	43 (42.2)	0.841
Resection, *n* (%)			1.000
R0	33 (97.1)	98 (96.1)	
R1	1 (2.9)	4 (3.9)	
Postoperative complication, *n* (%)	7 (20.6)	26 (25.5)	0.649
Type of complication, *n* (%)			0.103
General	0	12 (11.8)	
Surgical	5 (14.7)	5 (4.9)	
Liver-related	1 (2.9)	5 (4.9)	
Mixed	1 (2.9)	4 (3.9)	
Clavien-Dindo grade, *n* (%)			0.583
I	2 (5.9)	4 (3.9)	
II	0	8 (7.8)	
IIIa	4 (11.8)	10 (9.8)	
IIIb	1 (2.9)	3 (2.9)	
IVa	0	0	
IVb	0	0	
V	0	2 (1.5)	
Hospital stay (days)	11.5 ± 10.1	8.0 ±7.6	0.032
Mortality within 90 days, *n* (%)	0	2 (2.0)	1.000
Recurrence, *n* (%)	19 (55.9)	44 (43.1)	0.235
Intrahepatic			0.063
Extrahepatic	19 (55.9)	32 (31.4)	
Both	1 (2.9)	6 (5.9)	
Management of Recurrence, *n* (%)	0	4 (3.9)	
Re-do resection			0.001
TACE	3 (8.8)	4 (3.9)	
RFA	3 (8.8)	27 (26.5)	
Medical	3 (8.8)	2 (2.0)	
Surgery and RFA	1 (2.9)	5 (4.9)	
TACE and RFA	1 (2.9)	0	

* Group 1 patients who underwent liver resection before introduction of advanced approach in 2012, group 2 patients who underwent liver resection after introduction of advanced approach in 2012.

## Data Availability

We could not share data collected for our study to others. The confidentiality rules of hospital.

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
