# Peer review of "A Case-Matched Analysis of Laparoscopic Liver Resection for Hepatocellular Carcinoma Located in Posterosuperior Segments of the Liver According to Adaption of Developed Techniques"

_medicina, 2022, doi:10.3390/medicina58040543_

Round 1
Reviewer 1 Report
The suggestion were appearred in the text.

Author Response
On behalf of all co-authors, I would like to submit our revised manuscript titled “A Case-Matched Analysis of Laparoscopic Liver Resection for Hepatocellular Carcinoma Located in Posterosuperior Segments of the Liver According to Adaption of Developed Techniques” for publication in this journal.
We appreciate the time and efforts that you have dedicated to provide insightful feedback to improve this manuscript. The suggestions helped us to see our article from a different perspective. We have revised our manuscript extensively as suggested, and have provided point-by-point responses to the reviewers’ comments. The changes made in response to the individual comments are described.
We sincerely hope that the manuscript is now suitable for publication in this journal, and would be pleased to respond to any further queries regarding this submission.
We look forward to hearing from you.
Sincerely,
Jai Young Cho, MD, PhD, FACS
Department of Surgery, Seoul National University Bundang Hospital, 300 Gumi-dong, Bundang-gu, Seongnam-si, Gyeonggi-do, 463-707, Korea
Telephone: 82-31-787-7098; Fax: 82-31-787-4078; E-mail: [email protected]

Reviewer 2 Report
I read with great interest the manuscript by Kwon et al., it deals with a very interesting argument like LLR in PS segments. The research design is appropriate, even if a propensity score match should be based on all parameters included in the difficulty scores, not only size (tumor location, extent of resection). There are some issues to solve in the drafting, as well as English form, the discussion about results.
1- An extensive editing of English language and style is required, both in the abstract and in the main text. There are several formal errors to check and correct (peri-operative includes post-operative... in the abstract methods there is "from 2003 to 2001"... many repetitions ..."Trendelemburg" is not correctly written... and so on, many others). In the keywords there is "pancreatic adenocarcinoma": Why?
2- Conclusion are too strong according to methods and results, they should be less emphasized (according to the need of perspective studies , and despite the dishomogeneity of the cohorts for extent of resection and type of surgery).
In the abstract, what do you mean with DFS "time" ?
3- In the introduction, the second consensus conference was held in Morioka in 2014, non in Iwate. Furthermore, a further consensus conference took place in Southampton in 2018 and should be mentioned as well. The aims of this study must be explained: how the technical improvement and adaption have influenced outcomes.
4-In the Methods section, authors should explain which are exactly the differences between group 1 and group 2: only time? Or exactly what? Why exactly 2011 was chosen? In the definitions, it should be explained why parenchymal sparing hepatectomies were prefered (first sentence) and not anatomical resections. Similarly, they should explain why hepatectomies and trisectionectomies were included (only PS segments or all Major resections?). The decision to undergo RFA or TACE or surgery were discussed in a Multidiscipinary team? it should be staten. About Clavien Dindo classification the reference is missing. Finally, was the study conducted according to Strobe or stroccs guidelines? In the statistic session, you report only mean and SD for continues variables, as well as T test for comparison. Does it mean that all parameters have a normal distribution? If not, non parametric tests should have been performed. If yes, it must be clarified.
5- About results: the group 1 has more than 20% of RP sectionectomy, same for hepatectomies. Group 2 less than 8% for each of them, and mainly tumorectomies. How can you compare operation time and blood loss of so different transection lengths? It's normal that an hepatectomy takes more time than a tumorectomy. You should clearly discuss this limitation in the discussion, and clarify the point. The difference in operation time shouldn't be emphasized. About the difference in type of resection, you should also explain why you did not matched cases also according to type and extent of resection, even if these parameters are included in all difficulty scores.
6- In the discussion, you should discuss your specific results. About the differences in the 2 periods that you analyzed. And talk about limitations and interesting or original findings of your paper. ABout Pringle maneuver you should mention the recent propensity score matched analysis published in 2020 in the Nature group journal. The last paragraph about parenchymal sparing resections, is not clear. The conclusions paragraph must be more structured.
Table 3: the analysis was performed after matching? Please state
Author Response

(The authors gave the same response as above.)

Round 2
Reviewer 2 Report
I appreciated some revisions of the original manuscript. In particular quality of the abstract and of the conclusion paragraph has improved.
Anyway some minor issues still must be solved:
-English form and grammar errors, typo errors ("Trendelemburg", not "TrAndelemburg...and so on), in particular in the Discussion paragraph
-Within the discussion, the results of this manuscript are not discussed. You should mention your results and correlate them to the literature that you cite.
-Among the limitations, you should clearly cite the higher number of tumorectomies in the group with higher operation time.
